# Advances in the Current Understanding of the Mechanisms Governing the Acquisition of Castration-Resistant Prostate Cancer

**DOI:** 10.3390/cancers14153744

**Published:** 2022-07-31

**Authors:** Yifeng Mao, Gaowei Yang, Yingbang Li, Guowu Liang, Wangwang Xu, Mingqiu Hu

**Affiliations:** 1Department of Urology, The Second Affiliated Hospital of Bengbu Medical College, Bengbu 233041, China; yifengmao2009@163.com; 2Anhui Province Key Laboratory of Translational Cancer Research, Bengbu Medical University, Bengbu 233030, China; ygwshuishui@163.com (G.Y.); xww8372@163.com (W.X.); 3Department of Center of Science, Maoming People’s Hospital, Maoming 525000, China; liyingbang2012@163.com (Y.L.); gwliang123@126.com (G.L.); 4Department of Urology, Maoming People’s Hospital, Maoming 525000, China

**Keywords:** destructive resistance prostate cancer, androgen receptor, Wnt pathway, Hh pathway, ncRNAs

## Abstract

**Simple Summary:**

Castration-resistant prostate cancer (CRPC) is an advanced form of prostate cancer with a low survival rate, as the CRPC patients only survive for 9–13 months on average. In this narrative review, we first outline the most common androgen receptor (AR) receptor-related mechanisms, highlighting the important role of ARs in the development of CRPC. We also discuss the key importance of non-coding RNAs (ncRNAs) in this setting, including long ncRNAs and microRNAs. Overall, studies of the molecular biological mechanisms governing the CRPC will facilitate the development of appropriate targeted therapeutics, improving treatment options for the CRPC patients.

**Abstract:**

Despite aggressive treatment and androgen-deprivation therapy, most prostate cancer patients ultimately develop castration-resistant prostate cancer (CRPC), which is associated with high mortality rates. However, the mechanisms governing the development of CRPC are poorly understood, and androgen receptor (AR) signaling has been shown to be important in CRPC through AR gene mutations, gene overexpression, co-regulatory factors, AR shear variants, and androgen resynthesis. A growing number of non-AR pathways have also been shown to influence the CRPC progression, including the Wnt and Hh pathways. Moreover, non-coding RNAs have been identified as important regulators of the CRPC pathogenesis. The present review provides an overview of the relevant literature pertaining to the mechanisms governing the molecular acquisition of castration resistance in prostate cancer, providing a foundation for future, targeted therapeutic efforts.

## 1. Introduction

Prostate cancer is the second leading cause of cancer-related death among men worldwide [1]. In China, relative to the rates from before 1990, PCA incidence has risen by 98.21% while corresponding mortality rates have fallen by 3.82%. Consistently, PCA incidence continues to rise [2]. At the time of initial diagnosis, most of the PCA patients exhibit progressive or metastatic disease in China, with androgen-deprivation therapy (ADT) being the treatment of choice for most of the patients with metastatic PCA (European Society of Urology; Guidelines for the Treatment of Prostate Cancer, 2020) [3]. While initially highly effective, however, the median time that patients respond well to ADT is just 18–24 months, after which the patient will progress to develop castration-resistant prostate cancer (CRPC), which is generally defined based upon serum testosterone levels (<50 ng/dL or 1.7 nmol/L) and biochemical (prostate-specific antigen (PSA) levels increasing three times in a row within one week, with at least two of these increases being by more than 50% of the lowest levels, PSA > 2 ng/mL) or radiological (such as two or more new bone lesions in bone scans or soft tissue lesions based upon solid tumor response assessment criteria) evidence of progression after castration treatment [4]. The molecular mechanisms that underlie this acquisition of ADT resistance, however, remain incompletely understood, with recent work suggesting that a number of factors may ultimately contribute to the development of CRPC, as discussed at length below.

## 2. Androgen Receptor Pathways

The androgen receptor (AR) is a 110 kDa 919 amino acid nuclear receptor (NR) family member encoded on the X chromosome (q11-q12) (Figure 1A). The AR is composed of a central DNA-binding domain (DBD), and C-terminal ligand-binding domain (LBD), an N-terminal structural domain (NTD), and a hinge region linking the LBD and DBD [5,6,7,8]. The AR is encoded by eight exons (Figure 1), with exon 1 encoding the NTD, exons 2,3 encoding the DBD, and exons 4–8 encoding the LBD [9,10,11].

Several different mechanisms link AR to the development of CRPC, including AR upregulation, de novo androgen synthesis, co-regulatory factor activity, AR gene mutations, and altered splicing (Figure 2).

## 3. Classic Mechanisms of AR Pathway Enhancement

### 3.1. AR Overexpression

The AR signaling reactivation most commonly occurs at the level of gene amplification or protein upregulation, with up to 80% of the CRPC patients harboring high AR gene copy numbers, among whom 20–30% of patients exhibit high gene amplification levels [12,13,14]. Such amplification, however, is uncommon among PCA patients that have not undergone hormone therapy. In one fluorescence in situ hybridization (FISH) study, the researchers found AR amplifications to be absent in the benign prostatic hyperplasia (BPH) samples, present in just 2% of the primary PCA tumors, but present in 23.4% of the CRPC tumors [11,15]. At the genetic level, two-fold increases in the AR mRNA expression levels have been reported in the CRPC tumors [16]. The AR overexpression can additionally occur through the enhanced stabilization of AR mRNA or proteins, or through increased transduction rates mediated by heat-shock proteins (HSPs), with HSP40 and HSP70 having been shown to bind the AR NTD and to then interact with the AR LBD, thereby contributing to its overexpression [17,18].

### 3.2. De Novo Androgen Synthesis

The persistent de novo production of androgens within the CRPC tumors can additionally contribute to enhanced AR activation and the progression of hormone-refractory prostate tumors [17]. Even following ADT, dihydrotestosterone (DHT) levels in the prostate tissue remain at ~25% of baseline; these levels are sufficient to drive altered gene expression and tumor progression through tumor epithelial cell signaling [19,20,21,22,23,24,25]. Nishiyama et al. further conducted a Gleason score analysis of the CRPC patients in which they found that the low levels of DHT in these patients were sufficient to promote AR receptor activation and tumor progression [19,26,27].

When multiple steroids (*HSD3B2*, *AKR1C3*, *CYP17A1*, and *CYP11A1*) are present [28,29,30,31], adrenal androgen precursors can be used to promote DHT synthesis within tumors through the 5α-diketone pathway, leading to dehydroepiandrosterone (DHEA) and androstenedione conversion into DHT without any requirement for testosterone [32,33,34,35]. Abiraterone is a specific inhibitor of CYP17A1, and it was also the first drug approved to treat CRPC [32,36].

In recent work, 11-oxygenated androgen (11OHA4) was additionally identified as a critical source of intratumoral androgen production [25,37,38], as it can serve as a precursor for the production of the peripherally active androgens, 11KT and 11KDHT. Functionally, 11KT-mediated AR activation has been shown to be similar to that mediated by testosterone, with 11KT and 11KDHT binding to the AR receptors with an affinity comparable to that for testosterone and DHT, respectively [39,40,41,42]. Pretorius et al. consistently found that 1KT and 11KDHT were capable of driving cellular growth through the upregulation of the AR regulatory genes, including KLK3, TMPRSS2, and FKBP5 in the LNCaP and VCaP PCA cell lines [40]. In vitro conversion experiments have also shown that 11KT and 11KDHT can remain present in the LNCaP and VCaP cells for longer than testosterone and DHT [40,43].

### 3.3. AR Co-Regulatory Proteins

Over 180 such AR co-regulatory proteins have been identified to date, with both co-repressors and co-activators functioning in a synergistic manner to regulate AR transcription [44]. These proteins can modulate transcription, RNA splicing, and epigenetic regulatory mechanisms, including methylation, acetylation, phosphorylation, and ubiquitination, thereby shaping PCA development and progression [45,46,47,48,49,50]. Notably, these co-regulatory proteins can promote the sustained transcriptional activity of AR even under low levels of androgen availability.

The interaction of specific co-activating proteins with AR, including JMJD2C, LSD1, 37CBP/P300, p160/SRC, and SUV39H2, can drive enhanced AR activation and the concomitant upregulation of the AR-dependent genes to augment tumor growth. Consistently, the inhibition of these activators has been linked to reduced AR expression and PCA tumor growth in ex vivo analyses [51,52,53,54]. Askew et al. reported that SUV39H2 can function as a co-activator for AR that enhances its androgen-dependent transcriptional regulatory activity through interactions with MAGE-A11 and AR under androgen-deficient conditions [49]. Moreover, the *AEEB1* gene encoding the regulatory protein βArr1 was found to be upregulated in the CRPC tumor tissues, with the βArr1 deletion resulting in impaired PCA tumor growth, invasion, and metastatic progression in vitro and in murine model systems [55]. The AR co-repressors, in contrast, function to counteract the activity of the co-activator proteins [56,57]. Tan et al. reported that the CRPC tumor cells exhibited decreased CKβBP2/CRIF1 expression relative to the levels observed in androgen-dependent PCA tumor cells, with a corresponding increase in the expression of the co-activator STAT3, contributing to synergistic AR signaling enhancement [58]. As such, the downregulation of the co-repressors and the upregulation of the co-activators can spur the CRPC onset and progression [11].

## 4. Abnormal AR Pathway Activation

### 4.1. Mutation of AR

Between 10 and 30% of the CRPC patients harbor point mutations that alter the conformation of the AR protein [12,59]. The mutations are most commonly (~45%) present within the LBD of the AR (e.g., T878A, L702H, H875Y, F876L, and T877A), with the NTD and the DBD being progressively rarer sites for such mutations [60,61,62,63,64]. The metastatic PCA tumors have been shown to be more susceptible to the LBD mutations under low androgen conditions following the application of the AR antagonists, with mutations such as F877L contributing to the conversion of enzalutamide or apalutamide into AR-activating agonists [65,66]. Similarly, the W741L/C and ARH874Y/ART877A mutations can convert anti-androgenic flutamide and bicalutamide into agonistic agents [67,68,69]. These mutations also reduce ligand binding-specificity and increase sensitization to other steroid hormones [70]. Through these actions, and the conversion of specific anti-androgenic agents into AR pathway agonists, these point mutations can thus favor CRPC progression [71,72].

### 4.2. Altered AR Splicing

The AR splice variants (AR-Vs) were initially identified over two decades ago, and over 22 of such variants have been identified to date (AR-V1, 3, 7, 8, 9, 567, etc.) [73,74,75,76,77]. As compared to wild-type full-length AR (AR-FL), these AR-Vs are constitutively active owing to a lack of the AR LBD [76,77,78,79]. As the LBD is the target of many drugs used in the context of the ADR (including abiraterone, enzalutamide, apalutamide, and darolutamide) [80,81,82], the AR-V formation can occur in response to the ADT or low androgen levels, ultimately leading to the development of CRPC and associated disease progression.

The AR signaling can drive the onset of CRPC under the conditions of persistent ADT treatment. The AR-Vs are thus commonly expressed in the CRPC tumors as an alternative to AR-FL signaling, potentially functioning in a manner distinct from other AR isoforms [83,84,85,86,87,88]. The AR-V7, for example, has been identified as the most clinically relevant AR-V owing to the fact that it lacks an LBD, yet is capable of maintaining combinatorial activity under low androgen conditions. High expression levels for this variant have been observed in 75% of the patients with CRPC, despite being present in <1% of primary PCA cases [83,84,85,88,89]. The AR-V7, AR-V3, and AR-V9 have also recently been detected at high expression levels within the PCA cells following treatment with abiraterone, maintaining the AR signaling activity in an androgen-independent manner conducive to the CRPC growth [76,90,91,92,93].

The ARV-567es is a common AR-V that lacks LBD exons 5–7, while retaining the hinge region encoded by exon 4 that is involved in the receptor nuclear localization [86,94,95,96]. In contrast to AR-V7, this variant has only been detected in cases of advanced or malignant PCA, including CRPC [87,94,97,98]. Liu et al. generated transgenic mice in which the AR-V567es was placed under the control of the probasin (Pb) promoter, revealing significant increases in oncogenic K-RAS, FLI1, STK33, NF-κB, and β-linked protein signaling, suggesting that this AR-V can drive both tumor growth and the acquisition of castration resistance [99].

## 5. AR Bypass Pathway: Glucocorticoid Receptor Induction

The GR and AR are highly homologous steroid receptors, particularly within the DBD domain. As it harbors the same chromatin binding site, the GR can regulate the expression of a large number of the AR-specific genes [100,101,102]. Studies have shown that the GR can inhibit tumor proliferation in androgen-dependent PCA, and, conversely, in the CRPC, the GR can lead to tumor progression [103,104,105]. The androgen receptor antagonists can reduce the inhibitory effect of AR on the GR expression, increase the GR expression in PCA, promote cell survival and proliferation, and promote the expression of CRPC. Arora et al. found that the GR inhibitors can restore the sensitivity of patients to enzalutamide [106]. Hoshi et al. demonstrated that the GR-mediated upregulation of GLUT4 is associated with the progression of the CRPC [107]. In addition, Purayil et al. found that βArr1 can play a role in the progression of prostate cancer to CRPC by binding to the GR to initiate a mitogenic signaling cascade [108]. As such, GR can bypass the AR pathway and promote the development of CRPC.

## 6. DNA Repair Pathway

DNA damage repair is one of the most fundamental defense mechanisms that governs cell survival [109,110]. The defects and changes in DNA repair genes have been reported in prostate cancer, and these altered DNA repair pathways are closely related to the development, invasion, and progression of CRPC, with alterations in DNA repair genes being evident in 20–25% of the CRPC patients [111,112,113,114]. The SNPs in various DNA damage-repair pathway genes (e.g., BER, NER, MMR), have been shown to be associated with CRPC progression [114,115,116,117]. The most common mutations are located in the homologous recombination repair (HHR) genes, particularly BRCA2, with the deletion of RB1 frequently impairing the DNA repair and increasing castration resistance [111,118,119,120,121]. Using LNCaP and LAPC4 cells, Chakraborty et al. have demonstrated that CRCA2 can cause a castration-resistant phenotype, and that the shared deletion of BRCA2 and RB1 in prostate cancer cells promotes an epithelial cell-to-stromal cell transformation, which is closely related to an aggressive phenotype [118]. Additionally, the known tumor suppressor and serine/threonine kinase ATM (Ataxia Telangiectasia Mutated) is involved in DNA damage repair, with recent work having shown it to be involved in the development of CRPC [122,123]. Tang et al. determined that ATR inhibitors can lead to an IFN-mediated autocrine apoptotic response initiated by cGAS-STING in CRPC, promoting the apoptosis of CRPC cells [122]. These results thus strongly suggest that blocking DNA damage repair can inhibit the development of CRPC.

## 7. Non-AR Pathways

### 7.1. Hedgehog Pathway Signaling

The Hedgehog (Hh) pathway has increasingly been reported to be derepressed in PCA cells under therapeutic treatment, contributing to the progression of CRPC [124,125]. Within the CRPC cells, the higher levels of the Hh pathway ligands (Shh, Ihh, Dhh) promote Smoothed (Smo) derepression, leading to its activation and the consequent activation of Glid, which is a transcription factor [124,126,127,128]. This Hh pathway plays a central role in cancer stem-cell maintenance, promoting PCA stem cell development, which can contribute to both metastatic progression and chemoresistance [126,129,130,131]. Under conditions of low androgen availability, the abnormal GLI signaling within the epithelial cells contributes to higher levels of p63 expression, resulting in an increase in cancer stem cell activity, facilitating proliferation, metastasis, and resistance to therapeutic intervention [129,132,133]. The AR co-activators Cli1-3 have also recently been reported to promote the AR-V activation through cross-regulation, linking the AR and Hh pathways in a manner conducive to the development of CRPC [78,134]. For example, Clid2 can serve as an AR co-activator [78,135,136]. Lu et al. reported that the expression of GLI2 is essential for the PCA androgen-independent growth, using the mice bearing LNCaP xenograft tumors, with the knockdown of GLI2 in this system being sufficient to prevent the development of CRPC, while GLI2 re-expression reversed this phenotypic change [137]. GLI3 has also been reported to interact with AR, enriching the AR-dependent gene expression programs in a manner that favors PCA growth in a resistant manner [138].

### 7.2. PI3K-AKT-mTOR Pathway Signaling

A majority of the CRPC tumors exhibit the activation of the PI3K-AKT-mTOR pathway [138,139,140,141,142,143], which is a central signaling regulator of diverse biological processes and governs the balance between pro- and anti-apoptotic signaling [144,145,146,147]. The AKT activation, for example, can lead to the inactivation of pro-apoptotic Bad and the activation of anti-apoptotic Bcl-2 and Bcl-XL, contributing to enhanced apoptotic resistance [141,148,149,150]. The enhanced PI3K-AKT-mTOR pathway activity can partially compensate for the disrupted AR signaling in PCA cells under the conditions of ADT treatment, thereby largely obviating the need for androgens, contributing to the development of CRPC [141,142,143,146,147].

Under the ADT treatment or in the absence of androgens, reductions in the oncogenic PTEN expression contributed to PI3K activation and downstream AKT/mTOR activation that is evident in 63% of the CRPC tumors [143,144,151,152]. Under the conditions of AR inhibition or AR deficiency, the PCA tumors can downregulate FKBP5, leading to the downregulation of PHLPP, which negatively regulated PI3K-AKT-mTOR signaling; thus, engaging this signaling pathway and contributing to enhanced tumor cell proliferation and survival, facilitating the acquisition of androgen-independent growth characteristics [142,153,154,155]. The AR, PI3K-AKT-mTOR, mitogen-activated protein kinase (MAPK), and Wnt-signaling pathways have also been shown to synergistically engage with one another to promote the proliferation and therapeutic resistance of the PCA cells [156,157,158,159].

### 7.3. Wnt Pathway Activity

The altered Wnt pathway activity has been tied to malignant growth and progression in many tumor types [8,9,10,160,161,162,163]. Consistently, Wnt co-stimulation has been identified as a key driver of advanced PCA progression, with the Wnt pathway alterations being evident in 18% of the CRPC tumors [10,11,12,14,157,162,163,164,165]. Zi et al. similarly detected the Wnt pathway activation in CRPC through RNA-seq and microarray analyses [165], and the KEGG pathway-enrichment analyses have highlighted the upregulation of key Wnt pathway components in PCA, following ADT exposure [166]. The Wnt pathway is separated into the classical Wnt/β-linked protein pathway, as well as the non-classical β-linked protein independent pathway [167]. In low-androgen settings, studies have found that AR interacts with the classical pathway, promoting the Wnt pathway activation [161,164,168]. On the one hand, a regulatory mechanism exists for Wnt and AR such that an increase or decrease in one can cause a reciprocal decrease or increase in the other. Indeed, Lee et al. previously observed in vitro that the AR pathway activation in LNCaP cells was reduced following enzalutamide treatment or exposure to low androgen levels, while the Wnt/β-linked protein-responsive transcriptional activity was enhanced, leading to more robust tumor cell growth [169]. On the other hand, the Wnt can interact with AR. For example, the downstream effector β-linked protein has also been identified as a coactivator of AR that is integral to the AR transcriptional complex formation, thereby promoting the acquisition of castration resistance [162,163,164,166,169,170].

The non-classical Wnt pathway was also recently identified as an important regulator of the CRPC tumor cell growth [167]. For example, in their analysis of 1519 PCA samples, Sandsmark et al. determined that this non-classical pathway can drive tumor invasivity [171]. Mechanistically, Wnt5a is an important activator of the non-classical pathway signaling activity [171,172]. Through a series of tissue microarray and xenograft model studies, Lee et al. found Wnt5a to be capable of activating the ERK pathway activity, inducing the expression of CCL2 within androgen-sensitive cells and contributing to macrophage infiltration and the development of CRPC [173].

### 7.4. Non-Coding RNAs

The non-coding RNAs (ncRNAs) are a large class of RNAs that lack protein-coding capabilities, and are subdivided into microRNAs (miRNAs), small nuclear RNAs (snoRNAs), PIWI-interacting RNAs (piRNAs), and long ncRNAs (lncRNAs). These ncRNAs can play diverse roles as the regulators of processes such as differentiation, metabolic activity, and apoptosis [174,175,176]. The dysregulation of specific ncRNAs can contribute to the development and progression of PCA, with many studies having thus explored patterns of differential ncRNA expression during the different stages of PCA [175,176,177,178,179]. An increasing number of ncRNAs have been identified using high-throughput RNA sequencing (RNA-seq) technologies, playing important roles in the development of CRPC through various mechanisms [180]. Of these, ncRNAs, miRNAs, and lncRNAs are the best studies and offer great potential as biomarkers and therapeutic targets.

#### 7.4.1. miRNAs

As the first ncRNA subtype to have been identified as a driver of CRPC, the miRNAs are short 20–24 nucleotide RNAs that can bind to complementary 3′-untranslated region (UTR) sequences of target mRNAs, thereby suppressing their expression or promoting their degradation [181,182]. Over 20 miRNAs have been linked to the development of CRPC, governing the survival, migration, proliferation, and apoptosis of these tumor cells [181,183,184]. These include tumor-associated miRNAs, such as miR-32, miR-21 and miR-125b, that are upregulated in prostate tumors wherein they promote tumor suppressor-gene downregulation to drive oncogenic activity [184,185,186], as well as tumor suppressor miRNAs (miR-212, miR-135a, and miR-488*), which are downregulated in PCA tumors, leading to the induction of proto-oncogene expression [187,188,189].

Through a qPCR analysis, Das et al. determined that the overexpression of miR-4719 and miR-6756-5p was associated with significant reductions in IL-24 expression within cells, which is noteworthy, given that IL-24 can promote tumor cell death via multiple mechanisms [190]. The CRPC tumor tissues were found to exhibit significantly higher miR-3195, miR-3687, and miR-4417 expression levels in stem-loop qPCR analyses relative to the corresponding levels in primary PCA tumor tissues [177]. The altered miRNA expression is thus likely to play a key functional role in the context of the development of CRPC.

#### 7.4.2. LncRNAs

At over 200 nucleotides in length, the lncRNAs represent a distinct ncRNA class that nonetheless serve as important regulators of a range of physiological and pathological processes [191,192,193,194,195]. The altered lncRNA expression has been tied to abnormal tumor cell apoptotic, migratory, invasive, and proliferative activity, with several lncRNAs having been identified as key regulators of the acquisition of castration resistance in PCA [196,197]. KDM4A-AS1, for example, is a lncRNA that is upregulated in the CRPC cells, wherein it can promote AR/AR-V deubiquitination, thus protecting them against downregulation [198]. As such, KDM4A-AS1 can promote significantly enhanced tumor cell viability, migration, and proliferation in vitro as well as more robust tumor growth in vivo [198]. PCBP1-AS1 has also been shown to prevent ubiquitin-proteasome pathway-mediated AR/AR-V7 degradation through the binding to the NTD of these receptor proteins [199,200]. The regulatory activity of this lncRNA has been tied to the CRPC tumor cell resistance to enzalutamide in vitro and in vivo.

The interactions between lncRNAs and miRNAs have also recently been shown to be mechanistically important in CRPC [174,201,202]. For example, CCAT1 is capable of interacting with DDX5P68 to promote nuclear AR-mediated gene (UBE2C, prostate specific antigen) activation, thereby spurring the development of CRPC and its progression. This lncRNA can also compete with cytoplasmic miR-28-5p, thereby interfering with its ability to suppress tumor progression [203].

## 8. Conclusions

Prostate cancer remains among the leading causes of cancer-associated death in males, yet the mechanisms underlying the development of CRPC remain incompletely understood. While AR signaling clearly remains important in this pathological context, a growing body of evidence supports the importance of other mechanisms and pathways independent of the AR pathway as drivers of castration-resistant disease (Figure 2). This paper also provides a brief overview of the role of ncRNAs, which have been an area of growing research interest in recent years. The mechanistic roles of these AR and non-AR pathway mechanisms are also closely intertwined, leading to the coordinated regulation of PCA progression. As the studies discussed above suggest, ADT is likely to remain a primary treatment for the advanced forms of PCA, including CRPC, although the combination of this therapeutic strategy with other inhibitor compounds may confer additional survival benefits to treated patients. For example, the inhibitors such as AR vs., Gli3, and FKBP5 have been shown to improve the overall survival of patients in combination with ADT, although further clinical verification of these findings is warranted. In addition, suppressor molecules targeting miRNAs and lncRNAs, such as cat1, kdm4a-as1, and kdm4a-as1, also warrant further study and exploration of their clinical relevance.

## Figures and Tables

**Figure 1 cancers-14-03744-f001:**
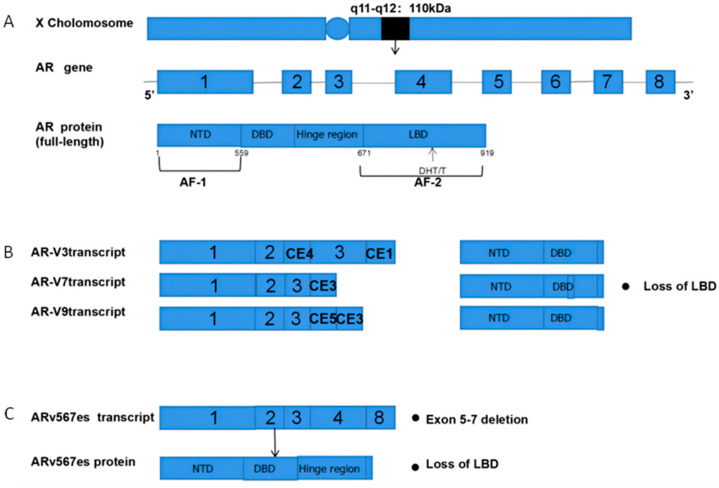
Structural overview of ARs and AR-Vs (AR-V3,7,9 and ARv567es). (**A**) Structural overview of the AR gene, located on the X chromosome q11-q12, encoding 919 amino acids and consisting of eight exons. The DHT and T ligands to the AR LBD; (**B**) The mechanisms underlying AR-V3, AR-V7, and AR-V9 production. Exons 4–8 are sheared to produce truncated AR-Vs that lack a LBD and Hinge region; (**C**) Mechanisms governing the production of ARv567es. Exons 5–7 are missing, resulting in truncated AR proteins and the lack of a LBD. AR = Androgen receptor; AR-Vs = AR variants; NTD  =  N-terminal transcriptional domain; DBD  =  DNA-binding domain; LBD  =  C-terminal ligand-binding domain; CE5  =  cryptic exon 5; CE3  =  cryptic exon 3; PAS  =  polyadenylation site; CE4  =  cryptic exon.

**Figure 2 cancers-14-03744-f002:**
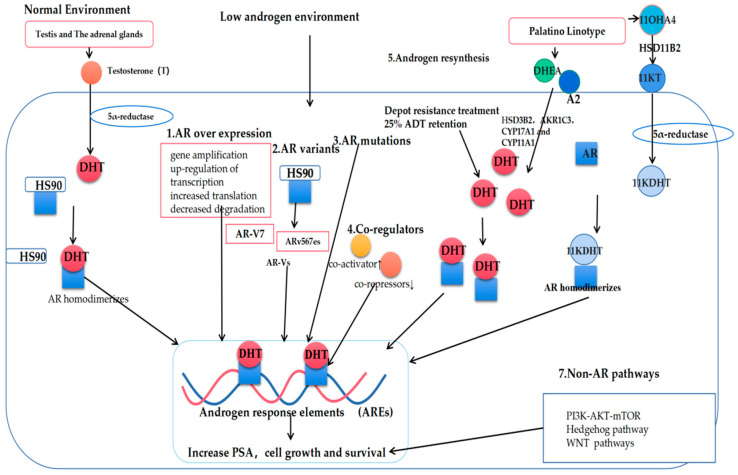
Mechanisms governing the progression of prostate cancer to castration-resistant prostate cancer (CRPC). T: Testosterone; DHT: Dihydrotestosterone; AR: Androgen receptor; AR-Vs: AR variants; DHEA: Dehydroepiandrosterone; A2: androstenedione; 11OXHA4: 11-oxygenated androgens; 11KT: 11-ketotestosterone; 11KDHT: 11-ketodihydrotestosterone.

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
