# Peer review of "Advances in the Current Understanding of the Mechanisms Governing the Acquisition of Castration-Resistant Prostate Cancer"

_cancers, 2022, doi:10.3390/cancers14153744_

Round 1

Reviewer 1 Report

This is the second revision. Authors have made clear effort to improve the manuscript. While I have some reservations on its structure and content, I see the authors’ position and agree the manuscript has updated the recent developments on AR and other pathways relevant to CRPC development. I support its publication.

Reviewer 2 Report

The concerns raised and suggestions given have been duly and completely addressed to my satisfaction.

This manuscript is a resubmission of an earlier submission. The following is a list of the peer review reports and author responses from that submission.

Round 1

Reviewer 1 Report

This review manuscript by Mao et al aims to update the developments on a set of classical processes or pathways with well-established contributions to CRPC development, including AR, Hedgehog (Hh) and PI3K-AKT-mTOR.  Non-coding RNAs was also briefly discussed. Nonetheless, its focus is on AR.

The relevance of AR, PI3K-AKT-mTOR, and Hh was well reviewed, particularly AR and PI3K-AKT-mTOR. This review largely represents the well-established knowledge without sufficient updates. The contributions of non-coding RNA to CRPC are an emerging concept, but this was only briefly discussed. As a result, this review does not add to the current literature.

The concepts covered in this review could be better developed. For instance, lines 228-243, it was not clear how low AR signaling strength leads to Wnt activation; was it strong AR signaling inhibiting Wnt or AR at its low signaling environment preferentially utilizes wnt?

Some statements could be more precise. Line 11: “Most prostate cancer patients eventually develop castration resistant prostate 11 cancer (CRPC) … …”. Line 33: PC is the 2nd leading cause of cancer death in men, not sure what was the evidence supporting the authors’ ranking of PC death at 10th among cancer cancer-caused fatality in men. Lines 37-38: it might need to indicate in which regions where PC was commonly diagnosed at advanced stages with metastasis, as this is unlikely the situation in many countries.

Reviewer 2 Report

The authors reviewed the current understanding oft he mechanisms of castration resistant prostate cancer. They included broad spectrum of literature in youre analysis.
The simple summery has some mistakes in the blank space and please capitalize at the beginning of the sentence.

The introduction is sufficiently. I would suggest that the searching words are listed in the instruction and which paper were  excluded in your review. However, Figure 2 is too small to read.
Please explain the AR Bypass pathways (Glucocorticoid Receptor Induction) and the DNA repair pathway.

Please explain the non-coding RNAs in more details and discussed are this markers clinically relevant.

Reviewer 3 Report

General comments:

1. Title:

     Advances in the current understanding of the mechanisms governing prostate cancer    

      acquisition of castration resistance

           Suggestion:

          Advances in the current understanding of the mechanisms governing acquisition of  

          Castration Resistance Prostate Cancer.

                                                  This title will better put the focus of the review on CRPC

2. Prostate specific antigen (PSA):

         PSA, as the main biomarker of prostate cancer, has significant clinical limitations.

 3. Conclusion:

           As the studies discussed above suggest, ADT is likely to remain a primary 307 treatment for advanced forms of PCA, including CRPC, although the combination of this 308 therapeutic strategy with other inhibitor compounds may confer additional survival 309 benefits to treated patients”.

         In such an elaborate review, suggesting/listing some/other inhibitor molecules that could be further investigated, would augment/justify the continuum of this current review.

4. References:

          The references are current enough and show/highlight the “current understanding” of the transition of PCA to CRPC.  Great job.

Round 2

Reviewer 1 Report

The authors have responded to all my specific comments, which I appreciate.

However, the main shortfalls of this review largely remain. The authors have added two sections: “4. AR bypass … …” and “5. DNA damage pathway”, aiming to dilute the AR content which has been extensively research and reviewed. By the way, should these be sections 5 and 6? Nonetheless, both sections were descriptive and lacked depth. This remains the pattern for all sections. Although Mao et al covered numerous processes relevant to CRPC development, all these processes were simply presented as concepts which were well known for their roles in CRPC requisition. It is hard for readers to appreciate the new developments in these processes and how these updates advance our current understanding of CRPC progression.

The AR content consists of at least 50% of this review. The concepts discussed are well known. Is it necessary to recap this knowledge to this extent? What are the new developments in the field? How may these developments contribute to our knowledge of AR in CRPC? Will this knowledge be relevant in management of CRPC?

Covering numerous processes also dilutes its (review) focus.